# Pulmonary Nodule Malignancy Classification Using its Temporal Evolution with Two-Stream 3D Convolutional Neural Networks

## Abstract

Nodule malignancy assessment is a complex, time-consuming and error-prone task. Current clinical practice requires measuring changes in size and density of the nodule at different time-points. State of the art solutions rely on 3D convolutional neural networks built on pulmonary nodules obtained from single CT scan per patient. In this work, we propose a two-stream 3D convolutional neural network that predicts malignancy by jointly analyzing two pulmonary nodule volumes from the same patient taken at different time-points. Best results achieve 77% of F1-score in test with an increment of 9% and 12% of F1-score with respect to the same network trained with images from a single time-point.

**Keywords:** Lung Cancer, Nodule Malignancy, Convolutional Neural Networks.

## 1. Introduction

Pulmonary nodule malignancy assessment done by radiologists is extremely useful for planning a preventive intervention, but it is a complex, time consuming and error-prone task. Current clinical criteria for assessing pulmonary nodule malignancy rely on visual comparison and diameter measurements of the initial and follow-up CT images (Larici et al., 2017). In this respect, three-dimensional assessment provides more accurate and precise nodule measurements (Ko et al., 2012). Despite the extensive literature on automatic classification of nodule malignancy (Dey et al., 2018; Causey et al., 2018; Ardila et al., 2019), to the best of our knowledge, there has not been any previously reported nodule malignancy classifier, trained with clinically validated nodule annotations, that analyzes more than one nodule image belonging to different CT scans of the same patient but taken at different time-points.

In this work, we take a step forward in this direction by building a novel nodule malignancy classifier, relying on 3D deep learning technologies, that incorporates the temporal evolution of the pulmonary nodules, trained on a longitudinal cohort of incidental nodules with clinically confirmed annotations at the nodule level.

## 2. Methods

We propose a nodule malignancy classifier using a two-stream 3D convolutional neural network (TS-3DCNN). Two-stream network architectures have been successfully applied in multitude of domains, specially for action recognition (Karpathy et al., 2014). This type of networks are composed of a feature extraction component in which two subnetworks (in our case, with shared architecture and weights) process a pair of images in parallel to produce two embedding feature vectors directly from the images. A second component, the

classification head of the network, generates a classification result (the nodule malignancy) from the two embedding feature arrays.

Currently, obtaining enough clinically confirmed annotated series of pulmonary nodules to properly train the TS-3DCNN from scratch is difficult. Therefore, we configured the sibling networks of the TS-3DCNN with a pre-trained 3D ResNet-34 network aimed at identifying pulmonary nodules from nodule candidates. The pre-trained network, following indications from (Bonavita et al., 2019), was trained using a large amount of nodule candidates (> 750K) from the LUNA-16 challenge (Setio et al., 2017) and reported competitive test performances (84.2% of F1-score).

Since it is difficult to know a priori which layer from the pre-trained network can provide the most informative features for our specific problem, we configured different TS-3DCNNs using different features maps (from the last layer of each of the 4 convolution blocks that form the pre-trained network). In accordance with the pre-trained network, the input of the TS-3DCNNs was a pair of patches of 32x32x32, cropped around the center of the annotated nodules at both time-points. All the patches were pre-processed before entering the network, by clipping their pixel intensities between -1200 and 600 HU and normalizing their values.

The classification head component of the TS-3DCNN was configured with a flatten, a concatenation, and a fully connected (FC) block layer. The FC block comprises a FC layer (with 64 units), a batch norm, a ReLU, a dropout and a final FC layer (with one unit). Figure-1 shows the architecture of the TS-3DCNN network.

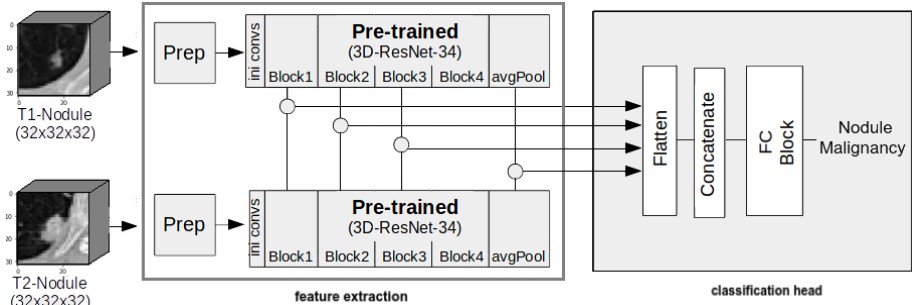

Figure 1: Two-stream 3D CNN for lung cancer classification.

To allow a fair comparison between the different TS-3DCNNs we defined the same initial training settings. Thus, binary cross-entropy was set as the loss function, the number of epochs was set to 150, the learning rate to 1e-4, the batch size to 32, dropout to 0.3, the early stopping to 10 epochs, and Adam was used for optimization. Moreover, random rotation and flip were applied for data augmentation.

## 3. Results

A dataset composed of 161 pairs of thoracic CT scans (103 cancer and 58 benign) were mostly collected at the beginning (T1) and at the end (T2) of the patient follow-up, that is, just before discharge (in the case of benign tumor) or surgical intervention, in the case of a malignant tumor. These data were obtained under institutional review board approval and were annotated by two different specialists. For each pair of CTs, the radiologists detected

and matched the most relevant nodule and annotated its malignancy. The nodules were labelled as malignant if they had a positive cancer biopsy, and benign if they did not have a significant change in their structure, density or morphology during 2 years or more.

To train the models, we used random stratified sampling to partition the data into training (70%) and testing sets. Also, we optimized the different models with the training data using a stratified 10-fold cross-validation.

| Model | Time | Train (F1) | Validation (F1) | Test | | |
|---|---|---|---|---|---|---|
| | | | | F1 | Prec | Recall |
| 3DCNN | T1 | $0.829 \pm 0.08$ | $0.853 \pm 0.04$ | 0.658 | 0.754 | 0.657 |
| 3DCNN | T2 | $0.871 \pm 0.04$ | $0.870 \pm 0.03$ | 0.686 | 0.782 | 0.650 |
| **TS-3DCNN** | **T1_T2** | **$0.875 \pm 0.08$** | **$0.869 \pm 0.07$** | **0.770** | **0.764** | **0.792** |

Table 1: Lung cancer performances of the configured experiments.

Table-1 shows the results of the best nodule malignancy classifiers using a single nodule image (T1 or T2) and using both (the TS-3DCNN approach). The best model was this last one obtaining a 0.77 of F1-score in test. This model outperformed by 9% and 12% the F1-score obtained by the best models using single time-point datasets. This result highlights the relevance of the nodule evolution for the malignancy classification. The best model using nodules from a single time-point obtained an F1-score of 0.68 at T2. This represents a 3% more of F1-score than the best model trained at T1. This result confirmed our intuitions since malignant nodules are usually easier to recognize when the disease is in a more advanced state, as they are bigger and more dense.

Recent studies (Causey et al., 2018) report nodule malignancy classifiers with high performances (approx. 86% specificity and 87% sensitivity). However, comparing these results with ours would be unfair. These classifiers have been trained using a public repository (Armato III et al., 2015) of single time-point nodule images, 10-times larger than ours and, especially when training deep-learning networks, this allows to build more accurate classifiers. In addition, we do not really know the complexity of the cases belonging to this repository, it could be that some of them are easy to diagnose, with clear symptoms of malignancy, and therefore, they would not require further studies. Our dataset consists of cases with at least a pair of CT scans taken before its medical diagnosis, evidencing their complexity. Finally, in the public repository, the annotations of malignity or benignity of the nodules are based on radiologists visual judgment. Our dataset, although smaller, has the advantage of relying on clinically validated annotations which is a more reliable way to evaluate the performance of the models.

## 4. Conclusions

We presented a two-stream 3D convolutional neural network model capable of predicting the malignancy of the pulmonary nodule using its temporal evolution. The best model obtained a F1-score of 0.77, which represents an improvement of approximately 12% and 9% of F1-score with respect to the best models using only a single nodule image at T1 and T2 respectively. A further extension of this work could involve collecting more cases and replacing the classification head with a recurrent neural network.

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
