# OpenReview forum: "Pulmonary Nodule Malignancy Classification Using its Temporal Evolution with Two-Stream 3D Convolutional Neural Networks"
_MIDL.io/2020/Conference — MIDL 2020_

### Official Review · AnonReviewer1 · 2020-03-13
**Nodule malignancy estimation from multiple timepoints using a two-stream CNN**

**Rating:** 3
**Confidence:** 5

**Review:**

This paper is well written and easy to follow. Malignancy estimation of pulmonary nodules is a relevant problem and indeed, growth is the most important risk predictor for cancer so using multiple scans is very relevant. In this paper, the authors proposed a two-stream CNN which takes two 32x32x32 volumes and has a classification head on top to produce a nodule malignancy. The authors nicely outlined how they trained the model and how the data was acquired. In the end, they show a substantial performance improvement over a single timepoint model.

Pros:
- Dataset with good reference standard set by pathology or 2 years of follow-up
- Good comparison with single timepoint models

Cons:
- Small dataset (30% of 161 cases means only 48 cases in the test set, of which approx 2/3 is malignant)
- It is not reported what the average size of the lesions at T1 and T2, and what the range and median of times between the two scans is. This is important information.
- No comparison with human performance on this dataset.
- No ROC analysis.

---

### Official Review · AnonReviewer2 · 2020-03-13
**weak accept with minor fixes**

**Rating:** 3
**Confidence:** 5

**Review:**

This paper proposes a pulmonary nodule malignancy classification based on the temporal evolution of 3D CT scans analyzed by 3D CNNs. It is an interesting idea and the quality is overall rather good for an abstract paper. Some points to address are listed in the following:

The early stopping is not clear. Specify that it is on the validation set if so, and clarify these points: “number of epochs was set to 150”, “early stopping to 10 epochs”

Why is this clipping used?

It is not clear whether T1 and T2 is available for all cases (“mostly”)

In Table 1, bold results are not always the best, this is very misleading.

It is strange that the T1, T2 generalize well to the validation set but not to the test. Can you comment?

“... obtained an F1-score of 0.68” -> 0.686?

---

### Official Review · AnonReviewer3 · 2020-03-14
**Merge the features from longitudinal CT scans for lung nodule malignancy classification**

**Rating:** 2
**Confidence:** 5

**Review:**

The authors proposed to utilize longitudinal scans for nodule malignancy classification. The proposed method is essentially applying the same backbone on two longitudinal CT scans and merge the feature vector for classification. The method was applied on an in-house dataset, and claimed not comparable with other dataset.

It’s meaningful to bring attention to longitudinal scans. The dataset is well-constructed.
However, the two-stream concept is not very solid in my opinion. For example, for the two-stream action recognition paper referenced, the two streams are spatial and temporal streams. Here it’s merely the same feature extractor and classification on the concatenated feature vector. The experimental comparison is not very meaningful. The main reading from the result is that longitudinal data is better than cross-sectional which is self-evident.

Detailed comments:
1) Can the authors comment on the gap between the training/validation F1 and the test F1. It seems the better performance of TS-3DCNN comes from better generalization capability.
2) Which blocks from CNN are finally used, or are all the blocks used as the figure suggests?

---

### Official Review · AnonReviewer4 · 2020-03-14
**Pulmonary Nodule Malignancy Classification Using its Temporal Evolution with Two-Stream 3D Convolutional Neural Networks**

**Rating:** 3
**Confidence:** 4

**Review:**

- Quality:
Interesting problem of predicting malignancy from longitudinal scans.
High-quality cohort of images, though small-sized.
Propose a "two-stream 3D convolutional neural network (TS-3DCNN)". Authors call it "sibling networks" rather than "siamese" architectures while sounding very similar.
Strong results on F-score improvement: "This model outperformed by 9% and 12% the F1-score obtained by the best models using single time-point datasets".
A confusion matrix would have been really informative, along with discussion on cases where prediction failed.


- Clarity:
"pair of patches of 32x32x32, cropped around the center of the annotated nodules at both time-points" & "the radiologists detected and matched the most relevant nodule and annotated its malignancy": while this is a quite impressive cohort and annotation work, sampling a single nodule per scan remains limited. Any lower threshold on size of nodule, as in [1]?

"The nodules were labelled as malignant if they had a positive cancer biopsy, and benign if they did not have a significant change in their structure, density or morphology during 2 years or more.": these are not exclusive conditions. What about a significant change and surgery but no confirmation of cancer via biopsy? Or vice versa, no change over 2 years but later on cancer?


- Originality:
This paper is very similar to [1] (some figures are identical) while the final task differs (matching distance versus nodule malignancy).

[1] Rafael-Palou X, Aubanell A, Bonavita I, Ceresa M, Piella G, Ribas V, Ballester MÁ. Re-Identification and Growth Detection of Pulmonary Nodules without Image Registration Using 3D Siamese Neural Networks. arXiv preprint arXiv:1912.10525. 2019 Dec 22.)

---

### Meta-Review · Area_Chair1 · 2020-04-04
**MetaReview of Paper94 by AreaChair1**

**Rating:** 4

**Metareview:**

The reviews as well as myself agree that studying longitudinal scans for nodule malignancy classification is interesting and valuable. The paper is well written and clearly presented.

**Paper Type:**

both

---

### Decision · Program_Chairs · 2020-04-11

Accept